# Metabolic Profiling of Pitaya (*Hylocereus polyrhizus*) during Fruit Development and Maturation

**DOI:** 10.3390/molecules24061114

**Published:** 2019-03-20

**Authors:** Yawei Wu, Juan Xu, Yizhong He, Meiyan Shi, Xiumei Han, Wenyun Li, Xingwu Zhang, Xiaopeng Wen

**Affiliations:** 1Key Laboratory of Plant Resource Conservation and Germplasm Innovation in Mountainous Region, Institute of Agro-Bioengineering/College of Life Sciences, Guizhou University, Guiyang 550025, Guizhou, China; yaweiwu2006@163.com; 2Institute of Pomology Science, Guizhou Academy of Agricultural Sciences, Guiyang 550006, Guizhou, China; xiaocao550100@163.com (X.H.); gzganju@163.com (W.L.); zzzzzxw@163.com (X.Z.); 3Key Laboratory of Horticultural Plant Biology, Huazhong Agricultural University, Wuhan 430070, Hubei, China; xujuan@mail.hzau.edu.cn (J.X.); shimeiyan@webmail.hzau.edu.cn (M.S.); 4Citrus Research Institute, Southwest University/National Citrus Engineering Research Center, Chongqing 400712, China; heyizhong@cric.cn

**Keywords:** pitaya (*Hylocereus polyrhizus*; ‘Zihonglong’), betalains, fruit ripening, metabolic profiling, biomarker metabolites

## Abstract

Pitaya (*Hylocereus polyrhizus*) has attracted much interest from consumers as it is a novelty fruit with high nutrient content and a tolerance to drought stress. As a group of attractive pigment- and health-promoting natural compounds, betalains represent a visual feature for pitaya fruit quality. However, little information on the correlation between betalains and relevant metabolites exists so far. Currently, color (Commission International del’Eclairage, CIE) parameters, betalain contents, and untargeted metabolic profiling (gas chromatography-time-of-flight-mass spectrometry, GC–MS and liquid chromatography tandem mass spectrometry, LC–MS) have been examined on ‘Zihonglong’ fruits at nine different developmental stages, and the variation character of the metabolite contents was simultaneously investigated between peel and pulp. Furthermore, principal component analysis (PCA) and partial least-squares discriminant analysis (PLS-DA) were used to explore metabolite profiles from the fruit samples. Our results demonstrated that the decrease of amino acid, accompanied by the increase of sugars and organic acid, might contribute to the formation of betalains. Notably, as one of four potential biomarker metabolites, citramalic acid might be related to betalain formation.

## 1. Introduction

Pitaya (*Hylocereus polyrhizus*), originating from Latin America and the West Indies [1], is an economically important fruit cultivated in tropical and subtropical regions. The pitaya fruit has antioxidant capacity due to the presence of betalains [2]. Betalains are a nitrogenous and water-soluble pigment which are classified as either red or crimson betacyanins, or yellow betaxanthins [3], and occur solely in the order of Caryophyllales [4]. As one of the major pigment classes, except for the base function of providing striking colors to plant organs for pollination and seed dispersal [3], betalains are good electron donors [5], natural food colorants [6,7], and have health-promoting properties as an antioxidant and due to their biological activities [8,9,10,11]. Their color properties can be utilized as chemical biosensors [12], protein-labeling fluorophores [13], and markers for genetic transformation [14,15]. Hence, betalains are becoming more attractive due to their physiological functions [7,10].

Pigments are organoleptic compounds that can combine with sugars, acids [10], and amino acids [16]. The accumulation of glycosylated pelargonidins and a small fraction of glycosylated cyanidins was a major contribution to receptacle pigmentation in most strawberry cultivars during ripening [17]. Furthermore, carbohydrates have a positive effect on anthocyanin biosynthesis in several plant seedlings [18]; for example, sugars, as signaling molecules, can regulate both flavonoid and anthocyanin contents in *Arabidopsis* seedlings [19] and pomegranate fruit [20]. Similar to anthocyanins, glycosylation reactions are also important in betalain biosynthesis [10]. As shown in Figure 1, betalains are tyrosine-derived pigments, and betalamic acid is the chromophore molecule of both betacyanins and betaxanthins, which condenses spontaneously with cyclo-DOPA to form betanidin, or with amino acids and other amines to form betaxanthins. Betanidin is further glucosylated by a betanidin glucosyltransferase to form the basic betacyanins, betanin or gomphrenin. An alternative pathway (dashed lines) occurs in *Mirabilis jalapa*, in which cyclo-DOPA is first glycosylated, then condensates with betalamic acid to form betanin [4]; eventually, betalains are stored in the vacuole as glycosides [21]. Sugars, amino acids, and amines are involved and play an important role in biosynthesis.

Betalains are involved in plant protection against abiotic and biotic stress [16], and also protect against certain oxidative stress-related disorders [22,23]. The study of betalains has also benefited from metabolomics, as around 75 betalains have been reported to date from 17 different plant families [24]. Suh et al. [1] investigated the different metabolites in red (*H. polyrhizus*) and white (*H. undatus*) pitayas, and found that most betalain-related metabolites were significantly higher in pitaya peel than in the flesh [1]. Recently, the primary metabolite profiles of three pitaya cultivars were investigated using GC-MS to provide fundamental information for making harvest decisions, and citramalic acid was identified, for the first time, in the pulp of the *Hylocereus* species [11]. Nonetheless, there is still much to be revealed about the metabolic mechanism of betalains. Nontargeted metabolomics is an efficient approach for exploring different metabolites, as well as to help elucidate the metabolic mechanism in the fruit-ripening process of many plant species, such as blueberries [25] and pepper [26]. However, information on the dynamic metabolite changes during pitaya fruit ripening is limited [11]; especially the correlation between betalains and relevant metabolites. In this study, we measured the color characteristics, betalain content, and metabolites of red pitaya fruits during different developmental stages. The major metabolites that contribute to betalain formation were further identified.

## 2. Results and Discussion

### 2.1. Fruit Coloration during Different Developmental Stages

As shown in Figure 2, both the pulp and peel of the mature ‘Zihonglong’ fruit (29 DPA, where DPA is days post-anthesis) were red, but the two parts were not synchronized; the pulp was colored earlier than the peel by about 2 days, as color-breaking in the pulp started at 26 DPA. Betalains formed quickly in the pitaya fruit, and the pulp color transformed from dotted or filamentous red pigment to fully red within 1 day (from 26 to 27 DPA), while the green faded in the peel at 27 DPA and turned completely red at 29 DPA.

The coloration parameters of the sampled fruit at each stage are shown in Table 1. In the peel, *L**, *b**, and *h°* values hardly changed from 19 to 27 DPA, but drastically decreased at 29 DPA. In the pulp, the *L** value gradually decreased from 78.25 to 32.64 as the fruit developed. The value of *b** decreased from 19 to 26 DPA, and then increased to maturity. The value of *h°* varied slightly, from 19 to 25 DPA (ranged from 73.91 to 92.82) and then decreased rapidly to fruit harvest (1.82). The *C** value decreased from 19 to 25 DPA, and then increased remarkably at 26 DPA in the pulp, but the *C** value slightly fluctuated (ranging from 22.45 to 29.33) from 19 to 29 DPA in the peel. The value of *a** was negative and increased gradually from 19 to 27 DPA, but shot up to 21.15 at 29 DPA in the peel. The values of *a** and *C** showed the same trend in the pulp, where they dropped to the lowest value (−0.21 and 4.41) at 25 DPA, and rose to peak value (29.18 and 29.37) at 27 DPA. Based on the variation features of the color parameters, lightness significantly decreased and was accompanied by the appearance of redness, and deepened in the peel and pulp, similarly to pomegranate [20,27]. The embedded seeds in the pulp may partially contribute to the appearance of blue color at 26 and 27 DPA, and darkening of the pitaya fruit pulp [27]. Consequently, the formation of betalains, degradation of chlorophyll, and darkening of thousands of seeds were major factors affecting fruit color characteristics as the fruit-ripening process advanced. In general, the peel and pulp of the ‘Zihonglong’ fruit were both red at the ripe stage, but pigmentation started earlier in the pulp than in the peel, which was similar to the results reported by Phebe et al. [27].

### 2.2. Betalain Contents

From young to mature fruit, the pitaya ‘Zihonglong’ fruit undergoes a rapid change in color, from green (in peel) or white (in pulp) to red-purple, due to the degradation of chlorophyll and the biosynthesis of betalains.

As shown in Figure 3, peel betacyanin content fluctuated slightly from 10 to 27 DPA (from 0.87 to 2.66 mg/100 g DW), and peaked at 10.43 mg/100 g DW, while betaxanthin content increased slightly from 10 to 27 DPA (2.37 to 5.59 mg/100 g DW), and increased significantly to 12.19 mg/100 g DW at 29 DPA. Betaxanthin content was higher than that of betacyanin throughout fruit development, while content differences between betacyanin and betaxanthin were significantly reduced at 29 DPA. In the pulp, betacyanin content was higher than that of betaxanthin at each stage, which was consistent with what was reported by Hua et al. [28]. Betacyanin and betaxanthin contents increased gradually from 10 to 26 DPA (3.40 and 2.70 mg/100 g DW, respectively), and then drastically increased to ripen the fruit. Notably, betacyanin attained 32.54 mg/100 g DW, which was 2.90-fold that of betaxanthin (11.21 mg/100 g DW) at the last stage. Suh et al. [1] reported that most betacyanins and betaxanthin were higher in the peel than in their pulp. In this study, betalain contents between peel and pulp were compared, and peel betaxanthin content was significantly higher than that of the pulp, whereas peel betacyanin content was less than that of pulp, and just one third of the pulp in the ripened fruit. The betalain contents were consistent with the Commission International del’Eclairage (CIE) parameters.

### 2.3. Compound Identification and Principal Component Analysis (PCA)

Metabolomics is an efficient approach to probe the correlation between phenotypes and metabolites [29]. Metabolic profiling of the fruit was investigated using an untargeted global metabolomics platform with GC–MS and LC–MS analysis. A total of 65 metabolites were identified, including 22 amino acids and amines, 16 sugars and sugar alcohols, 15 organic acids and fatty acids, 7 betalains and betalain precursors, and 5 other compounds (Appendix A).

PCA is usually used as the first step in chemometric analysis to visualize grouping trends and outliers [30]. In this case, PCA score plots were derived from nontargeted metabolite profiling and analyzed by GC–MS and LC–MS/MS. For PCA of primary metabolites from GC–MS in the pitaya peel and pulp, the first two components could explain 53.7% of metabolite variance. Component 1 explained 32.4% of the variance and Component 2 explained 21.3% (R2X = 0.654, Q2Y = 0.309; Figure 4A). Peel and pulp are clearly divided into two categories, and values for the peel and pulp were separated in the PCA score plot of pitaya fruit metabolites. PCA scores revealed that the metabolic composition of the pulp was different at each stage, while the metabolic composition of the peel was concentrated, indicating that metabolites in pulp tissue are more variable compared to peel tissue. For PCA of secondary metabolites from LC–MS in the pitaya peel (Figure 4B), the first two components were able to explain 36.7% of metabolite variance. Component 1 explained 21.8% of the variance and Component 2 explained 14.9%. For PCA of secondary metabolites from LC–MS in the pulp (Figure 4C), the first two components could explain 50.3% of metabolite variance. Component 1 explained 36.0% of the variance and Component 2 explained 14.3%. The PCA scores revealed that peel and pulp had relatively distinct metabolic profiles, and metabolites at different developmental stages were significantly different from each other. Hierarchical clustering analysis (HCA) of the metabolites was performed (Figure 4D). Peel and pulp were also clearly divided into two classes on the heat map, indicating significant differences in the metabolite content between peel and pulp. In addition, the stage-cluster feature was in agreement with the fruit developmental stage, indicating that the collected sample was reasonable, and that these metabolites could represent sample characteristics at each stage. The contents of amino acids, sugars, and organic acids in the pulp were higher than those in the peel, and amino acid content in the pulp decreased as fruit development progressed; conversely, the reverse fluctuation trend was observed for sugars and organic acids in the pulp, compared to amino acids.

### 2.4. Major Amino Acid and Secondary Metabolite Changes during Fruit Maturation

Amino acids and amines are responsible for fruit quality and nutritional values [26] and, as betaxanthin precursors, are important for better understanding of betalain biosynthesis [31]. To investigate the correlation between betalains and their relevant metabolites, comprehensive natural variation analysis of 20 amino acids and amines, and 10 secondary metabolites was carried out.

As illustrated in Figure 5A, in the peel, the concentrations of 15 amino acids and amines, alanine, serine, valine, proline, threonine, glutamic acid, glutamine, leucine, isoleucine, acetamide, arginine, methionine, phenylalanine, tyrosine, and tryptamine, fluctuated slightly throughout fruit development, and were lower than in pulp, whereas Suh et al. [1] reported that peel amino acid amount was relatively higher in comparison to the flesh; this discrepancy could be ascribed to differences in cultivar and environment. Furthermore, the condensation of betalamic acid with amino acids or their derivates leads to the formation of yellow betaxanthins, and amino acids are necessary for betaxanthin synthesis. We found that most amino acid contents in the peel were lower than in the pulp throughout fruit development, whereas the peel betaxanthin content was higher than that of the pulp; therefore, we deduced that amino acid content was not a limitation for betaxanthin formation. In the pulp, the contents of alanine, serine, glycine, valine, threonine, and arginine first increased and peaked at 16 DPA, followed by a constant decrease until fruit harvest. Proline and arginine first increased and peaked at 19 DPA, and then decreased until 27 DPA, when they subsequently showed a slight increase in the mature pulp. Glutamine, leucine, isoleucine, methionine, phenylalanine, and tyrosine decreased rapidly from 10 to 21 DPA, and then remained relatively stable until fruit harvest. The variation trend of amino acids in pitaya fruit pulp was consistent with what was reported in walnuts [32]. High levels of most amino acids were quantified at the early stages, and decreased with the development of pulp in pitaya, suggesting that these amino acids were transferred into sugars, organic acids, and secondary metabolites during development. Betalains are synthesized from tyrosine [10], and phenylalanine is a precursor of the phenylpropanoid biosynthetic pathway, which is related to the production of secondary metabolites in plants [26]. The decrease of tyrosine and phenylalanine may hence be attributed to the formation of betalains and other secondary metabolites. Asparagine is an amino acid which has the function of transporting nitrogen compounds in many plants, such as in white lupin (*Lupinus albus* L.); it accounts for 50%–70% of the nitrogen carried in translocatory channels serving fruit and seed [33]. Betalains are nitrogen-containing compounds [3], and the synthesis of betaine requires nitrogen. In the present case, a peak of aspartic acid content was synchronously detected at 25 DPA, both in the peel (0.20 mg/g DW) and pulp (0.24 mg/g DW). Therefore, the increase in aspartic acid could provide a nitrogen source for betalain synthesis. The concentration of γ-aminobutyric acid (GABA) in the peel was slightly higher than that in the pulp, but GABA concentration in both types of tissue demonstrated similar fluctuation trends, i.e., a tendency to gradually decrease. In peel, the tyramine concentration first decreased, then increased from 16 to 23 DPA, and then decreased again. Tryptophan concentrations in the peel hardly changed from 10 to 26 DPA, but increased rapidly in the following 3 days to fruit harvest, whereas tryptophan content in the pulp decreased from 10 to 21 DPA and then increased slightly until maturity.

For the variation of secondary metabolites (Figure 5B), the content of betanin, isobetanin, phyllocatin, and hylocerenin could not be quantified before 26 DPA, but later increased remarkably in the pulp. Isobetanin and hylocerenin were not detected in the peel, and betanin and phyllocatin were only detected in the peel of mature fruit, and their contents were far lower than in the pulp. The content of quercetin 3-*O*-rutinoside and cyclo-5-*O*-glucoside in the pulp were higher than that in the peel. Throughout the developmental stages, the concentration of quercetin 3-*O*-rutinoside in the pulp gradually decreased, while slightly increasing in the peel. Cyclo-5-*O*-glucoside concentration in the pulp rapidly decreased from 10 to 16 DPA, and then hardly varied until maturity; conversely, it slightly decreased in the peel before 21 DPA, and then fluctuated until maturity. No remarkable variation trend was investigated in the vebonol and sespendole concentration. Betalamic acid underwent condensation with an amino acid or amine to give rise to the formation of betaxanthin [34]. In this case, betalamic acid peaked in the pulp and decreased to a minimum in the peel at 26 DPA; the variance in levels of betalamic acid and betalains (such as betanin) took place during the same stage (26 DPA). Hence, we deduced that the significance of the variance of betalamic acid might be its involvement in betalain formation. Vapiprost content decreased in the pulp and slightly fluctuated in the peel with the development of the fruit. Metabolomics is a promising tool that has been successfully applied in the investigation of metabolism in plant development [17,35], and can also be used to identify new metabolites and quantify metabolic changes in diverse plant species [17,36]. Betalains and anthocyanins occur in a mutually exclusive fashion, whereas the anthocyanidin synthase (*ANS*) and dihydroflavonol 4-reductase (*DFR*) transcripts have also been identified in the tissues of betalain-producing plants, such as *Spinacia oleracea, Phytolacca americana, Bougainvillea glabra*, and *Mirabilis jalapa* [4]. Quercetin 3-*O*-rutinoside (a type of flavonoid) was identified and quantified in pitaya fruit herein. Other secondary metabolites, such as vebonol, sespendole, and vapiprost, were identified and reported, but their function in pitaya fruit development was not clear.

### 2.5. Changes of Major Sugars and Acids during Fruit Maturation

Soluble sugars were the anthocyanin donor substrates and acted as signaling molecules in the anthocyanin system [37]. Additionally, organic acids contribute to the stability of anthocyanins [38], and the positive correlation between organic acids and anthocyanins was reported [39]. Soluble sugars and organic acids also play an important role in the formation of fruit anthocyanin [20,37,40,41,42,43,44,45]. Hence, the contents of 12 sugars and sugar alcohols, and 9 organic acids at different pitaya fruit developmental stages were demonstrated, and their correlation with betalain formation was assessed.

Fructose, glucose, and sucrose are the three major components that contribute to the total sugar content in most ripe fruit [46]. As shown in Figure 6A, we found that the three predominant sugars detected were glucose (124.44 mg/g DW), fructose (113.03 mg/g DW), and sorbose (37.38 mg/g DW) in mature pitaya fruit pulp, and accounted for about 43.21%, 39.25%, and 12.98% of total sugar, respectively, whereas the concentration of sucrose, at 8.97 mg/g DW, was only 7.21% of the amount of glucose and 7.94% of the amount of fructose. Sugar composition and major sugars were consistent with values previously found in other pitaya varieties [1,11]. The concentration variation of fruit fructose, glucose, and sorbose hardly changed from 10 to 26 DPA (unripe fruit), but rapidly increased at 27 DPA (at the end of fruit development) until maturity in the pulp, which was in agreement with the available data for other pitaya varieties [11] and other fruit species, including apple, medlar, strawberry, grape, and wolfberry [47]. The variation features of glycoside and glucopyranose content were similar with those of glucose, fructose, and sorbose. Sucrose content hit its ceiling at 16 DPA, followed by a decrease until 23 DPA. After that, it increased steadily until maturity. The contents of myo-inositol ranged from 16.79 to 33.60 mg/g DW, which reached a peak at 16 DPA in the pulp, and then remained almost unchanged until fruit harvest. The contents of mannobiose, fructopyranose, and xylose decreased from 10 to 21 DPA; after that, mannobiose contents increased slightly until 26 DPA, followed by another decrease, while the contents of fructopyranose and xylose hardly changed from 21 DPA until fruit maturity. The content variation of both mannose and fucose showed a unimodal curve, with the peak appearing at 19 DPA (0.16 mg/g DW in mannose) and 23 DPA (0.31 mg/g DW in fucose), respectively. In the peel, fructose contents (ranging from 18.60 to 42.33 mg/g DW), glucose (ranging from 14.17 to 29.06 mg/g DW), sorbose (ranging from 2.74 to 6.17 mg/g DW), and glycoside (ranging from 0.29 to 0.75 mg/g DW) fluctuated slightly and were higher than those in the pulp from 10 to 26 DPA, and then rapidly decreased until maturity (0.95, 1.60, 0.72, and 0.03 mg/g DW, respectively). Sucrose and myo-inositol contents were lower than those of the pulp throughout fruit development; the former had two peaks at 16 DPA (5.31 mg/g DW) and 25 DPA (5.23 mg/g DW), and the latter ranged from 6.36 mg/g DW (29 DPA) to 12.50 mg/g DW (26 DPA). Fucose content peaked in the middle of fruit development and ultimately decreased, and there was a significant increase in mature fruit. The content variation of mannose showed a bimodal curve, and the same variation was observed with glucopyranose, fructopyranose, and xylose in the peel. Glycosylation reactions are frequent in betalain biosynthesis [10]; from 26 DPA to maturity, sugar and glycoside concentration in the pulp dramatically increased, and that in the peel significantly decreased. The remarkable formation of sugars and glycoside in the pulp may contribute to betalain production, whereas the decrease of major sugars in the peel suggested that the sugars and glycoside were used for the glycosylation reactions in betalain biosynthesis.

Nine organic acids were quantified in pitaya fruit (Figure 6B), namely, malic acid, citramalic acid, succinic acid, fumaric acid, citrus acid, quinic acid, 2-ketoglutaric acid, hexadecanoic acid, and oxalic acid. The organic acid content in the pulp was higher than that of the peel in mature fruit. In the mature pulp, the most abundant organic acid was malic acid, accounting for about 75% of total organic acids, while citramalic acid came second (at about 23% of the total organic acids). Malic acid content decreased from 10 to 21 DPA, then increased to a peak at 27 DPA in the pulp. In the peel, malic acid content was less than that in the pulp, but the variable trend was similar to the pulp; furthermore, malic acid content decreased to almost zero in the peel of the mature fruit. Citramalic acid was the predominant organic acid in unripe fruit and, interestingly, the contents in peel (ranging from 8.27 to 13.37 mg/g DW) and pulp (ranging from 5.56 to 13.13 mg/g DW) were higher than the malic acid content from 10 to 26 DPA. A similar result for citramalic acid was also reported by Hua et al. [11]. Citramalic acid is an analog of malic acid and can inhibit its accumulation [11]. Strangely, there were similar variation trends in the contents between citramalic acid and malic acid, reflecting no remarkably inhibitional performance. Succinic acid, citric acid, hexadecanoic acid, and oxalic acid content remained unchanged or varied slightly throughout fruit development in the peel. Citric acid content in the pulp dramatically decreased from 10 to 21 DPA, and then fluctuated slightly until maturity, which was different from that in mango fruit [48]. Concentrations of citramalic acid and succinic acid reached a high point in the peel at 26 DPA.

### 2.6. Analysis of Correlation Tests between Metabolites and Betalains

Significant positive correlations between anthocyanin and TA, citric acid, TSS, fructose, and glucose contents in pomegranate fruits [20], and significant negative correlations between UDP-galactose, succinic acid, acetic acid, and cyanidin 3-galactoside (cy3-gal) concentrations, and significant positive correlations between sucrose and cy3-gal were found in apple [39]. Here, correlation tests were performed between these metabolites and betalains. As shown in Table 2, in the pulp, there was a significant positive correlation between fructose, glucose, sorbose, glycoside, citric acid, and betacyanin content, as well as betaxanthin levels. Correlation tests demonstrated a negative correlation between most amino acids, citramalic acid, oxalic acid, myo-inositol, and betalains, whereas a positive correlation between malic acid, sucrose, and betalain contents was found in this study. In the peel, there was a negative correlation between sugars/sugar alcohols, organic acids, most amino acids, and betalains in the peel but, in this study, no significance was found. The result in our study is similar with a report on apple [39], that is, glycosylation reactions are an important process in betalain biosynthesis [10], and a synchronous relationship was found between betalain formation and an increase the concentration of major-sugars.

### 2.7. Identification of Representative Metabolites Related to Betalain Formation

The content of most metabolites demonstrated a remarkable variance in the pulp at 26 DPA (the stage of color-broken), suggesting that the color-broken stage was a crucial time point to study the coloring mechanism. Analysis of CIE parameters and betalain contents demonstrated that the period from 27 to 29 DPA was critical for betalain formation in the peel.

To discover which metabolites were related to betalain formation, partial least-squares discriminant analysis (PLS-DA) was performed on metabolites between peel-29 DPA (red peel) and peel-27 DPA (green peel), as well as metabolites between pulp-26 DPA (color-broken) and pulp-25 DPA (white pulp), between pulp-26 DPA (color-broken) and pulp-27 DPA (red pulp); metabolites with variable importance in projection (VIP) > 1.0 and *P* < 0.05 were selected as representative differential metabolites. PLS-DA clearly positioned a difference in development of the peel between 29 and 27 DPA (R2X = 0.666, R2Y = 0.999, Q2 = 0.955; Appendix A), so that the first two components could explain 66.58% of metabolite variance. Component 1 explained 35.11% of variance, and Component 2 explained 31.47% (Appendix A); 13 differential metabolites were identified (Table 3). A clear difference can be observed between pulp-26 DPA and pulp-25 DPA by PLS-DA (Appendix A) (R2X = 0.690, R2Y = 0.993, Q2 = 0.956). The first two components could explain 69.04% of metabolite variance. Component 1 explained 47.02% of variance, and Component 2 explained 22.02% (Appendix A); 13 differential metabolites were identified (Table 3). A clear difference can be observed between pulp-26 DPA and pulp-27 DPA by PLS-DA (Appendix A; R2X = 0.690, R2Y = 0.993, Q2 = 0.956); the first two components were able to explain 69.04% of metabolite variance. Component 1 explained 43.79% of the variance, and Component 2 explained 20.94% (Appendix A); 19 differential metabolites were identified (Table 3). Three representative differential metabolites, i.e., tyramine, citramalic acid, and tryptamine, were shared in both peel and pulp, and could be considered as representative differential metabolites for betalain formation. Interestingly, the citramalic acid level was also the highest at the color-breakdown stage and was identified as a representative differential metabolite linked to the presence of betalain pigments. It was postulated that citramalic acid participated in the TCA cycle, although no data supported this suggestion [49]. Potential functions of citramalic acid have rarely been reported. Noro et al. [50] suggested that citramalic acid is related to anthocyanin development in apple skin. The exudation of citramalic acid increased while under phosphorus deficiency in sugar beet roots, which enhanced the availability of solubilized soil phosphorus [51]. Information regarding the physiological function of citramalic acids in plants has been limited until now.

## 3. Conclusions

Our results demonstrated that betalain formation was consistent with CIE parameters. Betalain content increased dramatically during ripening. In the peel, betaxanthin content was higher than that of betacyanin at the nine stages, and vice versa in the pulp. The betaxanthin content of peel was higher than that of pulp, whereas peel betacyanin content was lower than that of pulp. Using GC–MS and LC–MS, 65 metabolites were detected in pitaya fruit, from which 51 major metabolites were quantified. The metabolic profiles of peel and pulp were effectively discriminated by PCA, and the stage-cluster feature was in agreement with fruit stage development. In general, the content of amino acids, soluble sugars, organic acids, and secondary metabolites in pulp was higher than in peel. The content of major amino acids in the pulp decreased during fruit development; however, the content of major sugars and organic acid increased in the mature fruit pulp. There was a significant positive correlation between fructose, glucose, sorbose, glycoside, citric acid, and betacyanin content, as well as betaxanthin levels in pitaya pulp, which suggested that sugar might benefit pitaya pulp formation during later development. Furthermore, this study is pioneering in reporting citramalic acid as a representative differential metabolite contributing to the formation of betalain pigments in pitaya fruit. Such information may facilitate a better understanding of betalain formation.

## 4. Materials and Methods

### 4.1. Fruit Material

Seven-year-old pitaya (*Hylocereus polyrhizus;* ‘Zihonglong’) trees were used in this study. These trees were grown in a 2.0 × 3.0 m space in Luodian County (25°25′N, 106°44′E; elevation, 398 m), Guizhou Province, China. The trees received the same horticultural practices, and disease and pest control. Flowers were pollinated on the same day as they were tagged. Fruits were sampled from 10 DPA (the seeds in pulp begin to form and fruit sampled can be determined whether is developing normally) to maturity (29 DPA) in 2015, including 9 different developmental stages. Thirty of the tagged fruits were randomly picked and divided into 3 groups with 3 biological replicates for each stage. Pitaya maturity was judged by visual observation of the fruit skin color changing to red. The color parameters of the collected samples were first measured, and the peel and pulp were then separated from the fruit and quickly cut into pieces, immediately frozen in liquid nitrogen, and mixed well, and samples were stored at −80 ℃ until analysis. Prior to total betalain determination, and GC–MS and LC–MS profiling, samples were lyophilized with a freeze-drier (SRK, German), and subsequently ground to a powder in liquid nitrogen.

### 4.2. Determination of Color Rating

Fruit peel and pulp color were measured using a colorimeter (CHROMA METER CR-400 Chroma Portable, Konica Minolta Sensing, Inc., Osaka, Japan). Color was assessed according to the Commission International del’Eclairage (CIE) and expressed as *L**, *a*,* and *b** color values. L* represented the relative lightness of colors, ranging from 0 (black) to 100 (white). *a** and *b** ranged from −60 to 60, where *a** was negative for green color and positive for red color, and *b** was negative for blue and positive for yellow [52,53]; chroma (*C** = (*a**^2^ + *b**^2^)^0.5^) and hue angle (*h°* = arc tan (*b**/*a**)) were determined as previously described by McGuire [52]. *C** expressed color saturation, with higher *C** values meaning less color type and a brighter color, while a lower C* value meant more color type and a bleak color. *h°* was expressed on a color wheel, where 0°/360° = red-purple, 90° = yellow, 180° = green, and 270° = blue [54].

### 4.3. Determination of Total Betalain Content

Betalains was measured according to Hua et al. [28] with a minor modification. About 0.1 g lyophilized peel and pulp were ground into powder in liquid nitrogen. Betalains were extracted with 1.8 mL of an 80% aqueous methanol (*v/v*) solution. After a 5 min vortexing, extracts were subjected to ultrasonic treatment for 30 min, followed by stabilization for 12 hours in darkness at room temperature. Subsequently, supernatants were collected at 10000*g* for 10 min, and residues were subjected to a similar second extraction. Supernatants were measured through a spectrophotometer, B = (A × DF × W × V × 100)/(ε × P × L), where B is betacyanin or betaxanthin content (mg/100 g of dried extract weight), A is absorbance (538 nm for betacyanins and 483 nm for betaxanthins), DF is the dilution factor at the moment of reading, W is the molecular weight (550 g/mol for betanin and 308 g/mol for indicaxanthin), V is the pigment solution volume (mL), ε is the molar extinction coefficient (60,000 L/mol·cm for betanin and 48,000 L/mol·cm for indicaxanthin), P is the fresh pigment weight (g), and L is the cell length (1 cm).

### 4.4. Metabolic Profiling

For primary metabolic profiling, each sample (0.2 g freeze-dried peel powder or 0.1 g freeze-dried pulp powder) was extracted with 2.7 mL of 80% methanol cooled at −20 °C and 300 μL internal standard (0.2 mg/mL ribitol in water). Nontargeted metabolite profiling was carried out by GC–MS (Thermo Fisher Scientific, Waltham, MA, USA) using a modified method as described by Sheng et al. [55].

For secondary metabolic profiling, about 0.1 g freeze-dried powder of peel or pulp was extracted with 80% methanol; the extraction process of secondary metabolites was the same as that described in Section 4.3. Secondary metabolic profiling was performed using a QTOF 6520 mass spectrometer (Agilent Technologies, Palo Alto, CA, USA) coupled with a 1200 series Rapid Resolution HPLC system, using the approach of Tan et al. [56]. Three replicates were used for each sample at the same stages.

The identification of alanine, proline, glycine, serine, threonine, aspartic acid, GABA, glutamic acid, glutamine, fructose, glucose, sucrose, sorbose, malic acid, citramalic acid, succinic acid, fumaric acid, citrus acid, quinic acid, tartaric acid, and oxalic acid was confirmed using standards, and the rest of the primary metabolite ions were identified by matching National Institute of Standards and Technology (NIST); a match quality of over 70% was accepted. Secondary metabolites and other amino acids were identified by comparing the characteristic fragment ion with a reported reference [1,57,58,59] and metabolites with similar fragment ions were suggested to be the same compounds.

### 4.5. Multivariate Statistical Analysis

PCA, PLS-DA, and the identification of representative differential metabolites (metabolites with VIP > 1.0 and *P* < 0.05) were performed with SIMCA-P + 11.0 (version 11.0, Umerics, Umea, Sweden) software. Metabolite data were first row z-score-normalized, then heat-mapped. HCA was performed using Mev (MultiExperiment Viewer) 4.9 software. Color parameter and compound content data were processed with Excel (Microsoft, Seattle, WA); then, one-way ANOVA was employed for statistical analysis of betalain, amino acid, sugar, organic acid, and secondary metabolite concentrations, as well as for significant analysis of color parameters, followed by a Duncan’s multiple range test at the 5% level (*p* ≤ 0.05) in SPSS17.0 (SPSS Inc., Chicago, IL). All data are expressed as means ± standard deviation of 3 replicates.

## Figures and Tables

**Figure 1 molecules-24-01114-f001:**
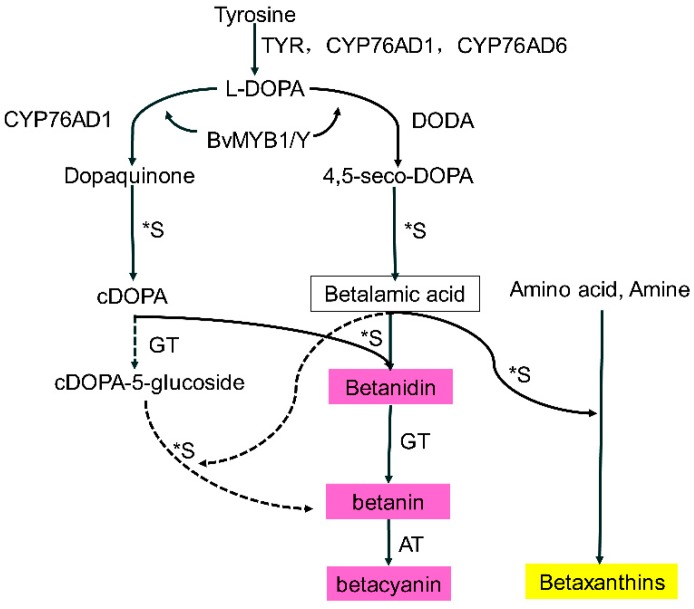
Betalain biosynthetic pathway (drawn according to [4]); TYR, tyrosinase; CYP76AD1, cytochrome P450; DODA, DOPA 4, 5-dioxygenase; BvMYB1/Y; *S, spontaneous steps; GT, glycosyl transferase; AT, acyltransferase; DOPA, 5, 6-dihydroxy-phenylalanine.

**Figure 2 molecules-24-01114-f002:**
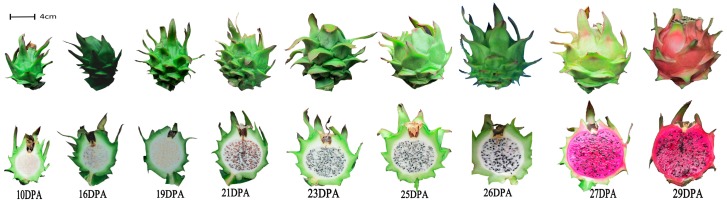
Experimental design of pitaya ‘Zihonglong’ (*Hylocereus polyrhizus*) harvested at nine stages of development.

**Figure 3 molecules-24-01114-f003:**
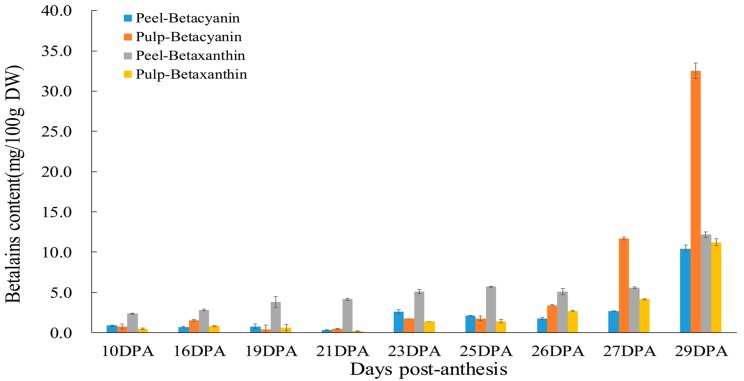
Betacyanin and betaxanthin contents during different fruit-developmental stages.

**Figure 4 molecules-24-01114-f004:**
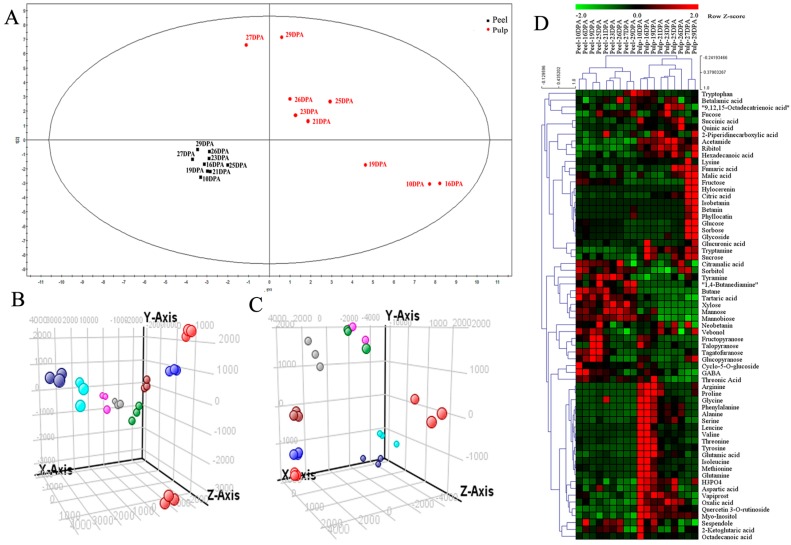
Principal component analysis (PCA) score plots and metabolite hot map derived from nontargeted metabolome profiling. (**A**) Metabolite PCA from the GC–MS of fruit peel and pulp; metabolite PCA from the LC−MS of fruit (**B**) peel and (**C**) pulp (bright-red dots, 10 DPA; blue dots, 16 DPA; red-brown dots, 19 DPA; gray dots, 21 DPA; green dots, 23 DPA; pink dots, 25 DPA; dark-red dots, 26 DPA; bright-blue dots, 27 DPA; dark-blue dots, 29 DPA); (**D**) metabolite heat map from GC–MS and LC–MS/MS.

**Figure 5 molecules-24-01114-f005:**
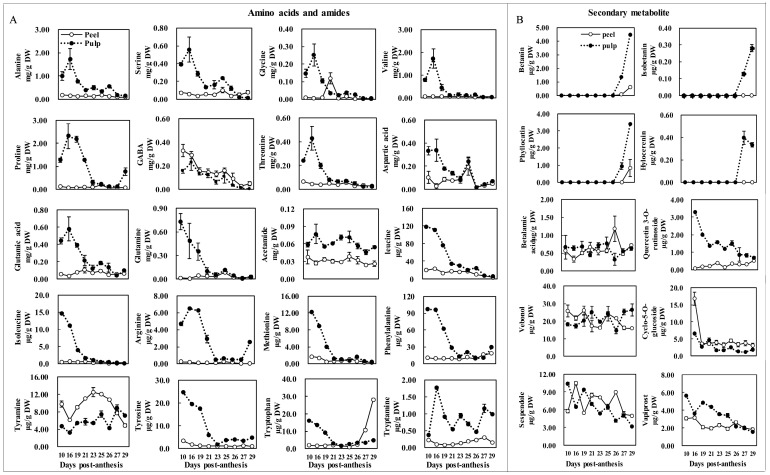
Developmental concentration of amino acids (**A**) and secondary metabolites (**B**) in pitaya fruit.

**Figure 6 molecules-24-01114-f006:**
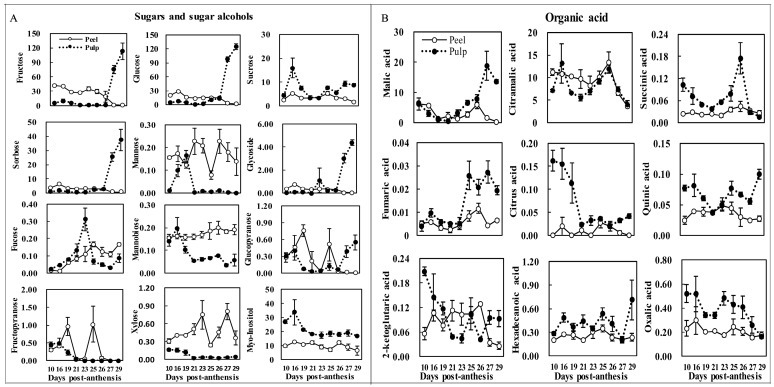
Developmental concentration of major sugars and organic acid in pitaya fruit. Note: The unit of sugar and acid content in the figure is mg/g DW.

**Table 1 molecules-24-01114-t001:** Coloration parameters of pitaya fruit at different developmental stages.

Tissue	Color Items	Harvest Date						
		19 DPA	21 DPA	23 DPA	25 DPA	26 DPA	27 DPA	29 DPA
Peel	*L**	47.91 ± 2.89	47.73 ± 2.36	43.30 ± 1.08	42.98 ± 1.47	43.24 ± 1.13	45.19 ± 0.24	37.69 ± 0.92
	*a**	−14.42 ± 0.49	−15.33 ± 0.31	−14.81 ± 0.31	−13.78 ± 0.36	−9.26 ± 0.42	−4.46 ± 4.47	21.15 ± 3.19
	*b**	22.36 ± 1.11	25.00 ± 1.12	22.34 ± 0.62	21.16 ± 0.98	21.33 ± 0.96	21.12 ± 0.89	10.38 ± 0.60
	*C**	26.60 ± 1.20	29.33 ± 1.11	26.80 ± 0.67	25.25 ± 0.96	23.44 ± 1.02	22.45 ± 1.92	23.64 ± 2.94
	*h°*	122.86 ± 0.42	121.55 ± 0.68	123.55 ± 0.33	123.11 ± 0.42	115.66 ± 1.03	100.04 ± 11.19	26.83 ± 3.27
Pulp	*L**	78.25 ± 1.26	67.69 ± 0.84	66.54 ± 3.21	66.10 ± 1.04	51.32 ± 1.65	38.24 ± 0.43	32.64 ± 0.84
	*a**	2.15 ± 0.44	2.38 ± 0.23	0.53 ± 0.87	−0.21 ± 0.96	14.38 ± 0.33	29.18 ± 0.68	27.41 ± 0.97
	*b**	15.18 ± 1.16	10.32 ± 4.05	6.88 ± 2.94	4.40 ± 0.12	−6.43 ± 0.14	−3.22 ± 0.60	0.60 ± 0.52
	*C**	15.35 ± 1.09	10.66 ± 3.96	6.96 ± 2.30	4.41 ± 0.12	16.68 ± 0.32	29.37 ± 0.73	27.42 ± 0.97
	*h°*	81.65 ± 2.14	73.91 ± 4.47	89.87 ± 5.04	92.82 ± 1.34	53.64 ± 2.15	6.26 ± 1.03	1.82 ± 0.44

**Table 2 molecules-24-01114-t002:** Correlation coefficients between betalains with metabolites during apple ripening.

Metabolites		Pulp		Peel	
		Betacyanin	Betaxanthin	Betacyanin	Betaxanthin
Amino Acids	Proline	−0.237	−0.311	−0.214	−0.162
	Glycine	−0.456	−0.502	−0.298	−0.172
	Valine	−0.365	−0.407	−0.456	−0.462
	Serine	−0.631	−0.657	0.321	0.297
	Threonine	−0.474	−0.522	−0.499	−0.568
	Aspartic Acid	−0.446	−0.504	−0.151	−0.095
	Glutamic Acid	−0.505	−0.553	−0.103	0.075
	Glutamine	−0.401	−0.452	−0.063	0.086
Organic Acids	Oxalic Acid	−0.803	−0.770	−0.446	−0.508
	Citramalic Acid	−0.511	−0.417	−0.808	−0.785
	Malic Acid	0.681	0.691	−0.504	−0.597
	Citric Acid	0.826 *	0.822 *	−0.367	−0.256
Sugars/Sugar Alcohols	Fructose	0.952 **	0.923 **	−0.681	−0.769
	Glucose	0.928 **	0.910 **	−0.683	−0.766
	Sorbose	0.954 **	0.934 **	−0.655	−0.740
	Glycoside	0.935 **	0.920 **	−0.622	−0.700
	Myo-Inositol	−0.402	−0.455	−0.709	−0.696
	Sucrose	0.375	0.356	−0.560	−0.501

Note: * indicates *p* < 0.05. ** indicates *p* < 0.01.

**Table 3 molecules-24-01114-t003:** Compounds determined to be of variable importance in projection through partial least-squares discriminant analysis (PLS-DA) on the identified compounds from GC–MS and LC–MS.

	Peel–29 DPA vs. Peel–27 DPA			Pulp–26 DPA vs. Pulp–25 DPA			Pulp–26 DPA vs. Pulp–27 DPA		
	Var ID (Primary)	VIP	*P*-Value	Var ID (Primary)	VIP	*P*-Value	Var ID (Primary)	VIP	*P*-Value
1	Tryptophan	1.696	0.0005	Arginine	1.476	0.0025	Betanin	1.374	0.0028
2	Betanin	1.670	0.0109	Phenylalanine	1.45	0.0488	Isobetanin	1.363	0.0066
3	Malic Acid	1.669	0.0026	Methionine	1.431	0.0208	Propanoic Acid	1.355	0.0005
4	Quercetin	1.663	0.0001	Aspartic Acid	1.413	0.0268	Aspartic Acid	1.353	0.0024
5	Tryptamine *	1.656	0.0154	Vapiprost	1.402	0.0143	Fructose	1.345	0.0135
6	Citramalic Acid *	1.620	0.0269	Sespendole	1.388	0.0183	Methionine	1.339	0.0168
7	Tyramine *	1.611	0.0046	Tyramine *	1.360	0.0482	Alanine	1.337	0.0045
8	Tyrosine	1.604	0.0120	Serine	1.353	0.0224	Valine	1.331	0.0055
9	Fumaric Acid	1.598	0.0060	Tryptamine *	1.326	0.0351	Hylocerenin	1.326	0.0208
10	Isoleucine	1.586	0.0147	Alanine	1.311	0.0269	Tyramine *	1.325	0.0210
11	Sucrose	1.523	0.0292	Glutamine	1.302	0.0369	Glucose	1.323	0.0235
12	Betalamic Acid	1.466	0.0439	Tryptophan	1.265	0.0393	Glycoside	1.322	0.0221
13	Fucose	1.448	0.0497	Citramalic Acid *	1.067	0.0306	Isoleucine	1.311	0.0272
14	NA	NA	NA	NA	NA	NA	Tryptamine *	1.298	0.0322
15	NA	NA	NA	NA	NA	NA	Citric Acid	1.295	0.0004
16	NA	NA	NA	NA	NA	NA	Mannose	1.283	0.0376
17	NA	NA	NA	NA	NA	NA	Phyllocatin	1.274	0.0414
18	NA	NA	NA	NA	NA	NA	Citramalic Acid *	1.239	0.0160
19	NA	NA	NA	NA	NA	NA	GABA	1.168	0.0342

Note: VIP, variable importance in the projection. Compounds accompanied with * are representative differential metabolites identified in both the peel and pulp. NA represents no data.

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
