# Peer review of "Metabolic Profiling of Pitaya (Hylocereus polyrhizus) during Fruit Development and Maturation"

_molecules, 2019, doi:10.3390/molecules24061114_

Round 1

Reviewer 1 Report

Just suggest the abstract should be made a little bit change, for example, the first sentence, Pitaya has brought interesting to costumers as its fruit novelty with high nutrient et al. Suggest to check and make some logically change.  

Author Response

Dear reviewer

Firstly, I would like to express our sincere thanks to you for the constructive and positive comments.

Point 1: Just suggest the abstract should be made a little bit change, for example, the first sentence, Pitaya has brought interesting to costumers as its fruit novelty with high nutrient et al. Suggest to check and make some logically change.

Response 1: Thank you for your precious comments on my paper, the abstract of the article has been improved by authors and English editing service department.

With best wishes,

Yours sincerely,

Xiaopeng Wen

Institute of Agro-biotechnology,

Guizhou University, Guiyang 550025,

Guizhou Province, P R China

Reviewer 2 Report

This manuscript describes metabolic profiling in the peel and pulp tissues of Pitaya with a focus on betalains. The manuscript needs to be substantially edited for English improved for readability. I have a few concerns/comments as listed below:

The manuscript presents metabolite profiling from 10 DPA. It is unclear why this stage was chosen. Why were earlier stages not sampled. This would have provided more comprehensive developmental information.

Many of the figures are not of good quality to visually assess the accuracy of the statements in the manuscript. This is particularly the case for Fig. 4. The fonts and symbol sizes need to be enhanced. The differences between stages and tissues is based on these analyses and it is important to be able to see the data clearly on this aspect.

One of main conclusions of the manuscript is the identification of citrmalic acid as a main component of pitaya fruit. However, this was already reported in reference 11 and also acknowledged by the authors. It was identified as a differential metabolite, but the biochemical significance of this is not discussed.

For the PLS-DA analysis, it is unclear to me as to why for the pulp, a comparison between 25 and 26 DPA was chosen (likely based on Table 1 data). The pulp did not change color visibly during this period as much as it did between 26 and 27 DPA (Fig. 2). Also, the betalain conc. change was more substantial between 26 and 27 DPA. This would have made a better comparison.

Changes in sugars/sugar alcohols and amino acids are primarily discussed in reference to the accumulation of betalains. However, these metabolites clearly have other major functions in fruit metabolism and growth. The developmental changes in these metabolites are therefore more likely to be associated with growth and developmental transitions.

Author Response

Dear reviewer

Firstly, I would like to express our sincere thanks to you for the constructive and positive comments.

Point 1: The manuscript presents metabolite profiling from 10 DPA. Why were earlier stages not sampled. This would have provided more comprehensive developmental information.  

Response 1: The original intention of our experiment was to study batalain. In view of the formation time of betalain in fruit pulp is general at 20DPA later, in addition, the seeds in pulp begin to form and fruit sampled can be determined whether is developing normally at about 10 DPA. Therefore, we conducted metabolite profiling from 10 DPA. It is true that the submitted manuscript did not explain clearly the basis for the selection of the sampling time, the content was supplemented in revised manuscript (The revised details can be found in Line 379-380, page 12).

Point 2: Many of the figures are not of good quality to visually assess the accuracy of the statements in the manuscript. This is particularly the case for Fig. 4. The fonts and symbol sizes need to be enhanced. The differences between stages and tissues is based on these analyses and it is important to be able to see the data clearly on this aspect.

Response 2: We pasted images directly into the submitted manuscript and resulted in the blurry of image sharpness. As for this problem, we processed image text more clear (The revised details can be found in Figure4, Line 163, page 5; Figure5, Line 236, page 7; Figure6, Line 288, page 9), and if revision is acceptable, we will upload TIFF images with 600 dpi to the editorial department.

Point 3: One of main conclusions of the manuscript is the identification of citrmalic acid as a main component of pitaya fruit. However, this was already reported in reference 11 and also acknowledged by the authors. It was identified as a differential metabolite, but the biochemical significance of this is not discussed.

Response 3: Though the physiological function of citramalic acids in plants is limited up to now, we have supplemented limited physiological functions of citramalic acid in plants(The revised details can be found in Line 344-348, page 10-11).

Point 4: For the PLS-DA analysis, it is unclear to me as to why for the pulp, a comparison. between 25 and 26 DPA was chosen (likely based on Table 1 data). The pulp did not change color visibly during this period as much as it did between 26 and 27 DPA (Fig. 2). Also, the betalain conc change was more substantial between 26 and 27 DPA. This would have made a better comparison.

Response 4: Thank you for your good suggestion, the comparison of differential metabolite between Pulp-26DPA and Pulp-27DPA was supplemented in Table 3, Line 351, page 11, and the relevant content was improved.

Point 5: Changes in sugars/sugar alcohols and amino acids are primarily discussed in reference to the accumulation of betalains. However, these metabolites clearly have other major functions in fruit metabolism and growth. The developmental changes in these metabolites are therefore more likely to be associated with growth and developmental transitions.

Response 5: Indeed, these metabolites are more correlative to fruit growth and developmental transitions, in view of our focus on the relationship between these metabolites and betalain formation, the association between metabolites and other fruit development process were few analyzed in the manuscript. For the enlightenment of your suggestion, the content of the correlation analysis between amino acid, sugar, acid and betalain in pitaya fruit was supplemented to further explore the relationship between them (that can be found in Table 2, Line 316, page 9).

Once again, thank you for your precious advice, I have found my shortcomings in my current work. I will follow your advice to improve my scientific research level in the future.

With best wishes,

Yours sincerely,

Xiaopeng Wen

Institute of Agro-biotechnology,

Guizhou University, Guiyang 550025,

Reviewer 3 Report

The paper, entitled "Metabolic profiling of pitaya (Hylocereus polyrhizus) during fruit development and maturation" is interesting. It reports the correlation between the betalains and different metabolites - amino acids, soluble sugars, organic acids, fatty acids.

I accept the paper for publication after minor revision concerning some following points.

In the introduction the authors present the plant and the importance of the produced pigments. However, are the authors sure of the species Hylocereus polyrhizus? H. polyrhizus and H. undatus, an other species, have the same red fruit skin color at mature stage but different flesh colors, red and white respectively. In the same way the cultivar "Zihonglong" is often described in litterature as a cultivar from H. undatus.

In the part Results and discussion, figures 5 and 6 must be improved. It is very difficult to read the axes.

The authors report the presence of citramalic acid as a a metabolite contributing to the formation of betalains. Can the authors present a putative biosynthetic way for the formation of this metabolite and from this metabolite to the betalains.

lines 101 and 111, the authors report that the color changing is due to a degradation of chlorophyll. Can the authors explain this degradation? What is the biosynthetic way between chlorophyll and betalains. Did the authors analyse the different chlorophyll a and b, but also xanthophylls. 

Author Response

Dear reviewer

Firstly, I would like to express our sincere thanks to you for the constructive and positive comments.

Point 1: In the introduction the authors present the plant and the importance of the produced pigments. However, are the authors sure of the species Hylocereus polyrhizus? H. polyrhizus and H. undatus, another species, have the same red fruit skin color at mature stage but different flesh colors, red and white respectively. In the same way the cultivar "Zihonglong" is often described in litterature as a cultivar from H. undatus.

Response 1: Thank you for your good suggestion. Indeed, Zihonglong was previously regarded as belonging to Hylocereus undatus (such as article written by Nie et al. Gene, 2015, https://doi.org/10.1016/j.gene.2015.03.007), however, international literatures classifies pitaya type with red flesh as Hylocereus polyrhizus. Therefore, the accurate Latin name of ‘Zihonglong’ should be Hylocereus polyrhizus, and we use this presentation herein.

Point 2: In the part Results and discussion, figures 5 and 6 must be improved. It is very difficult to read the axes.

Response 2: We pasted images directly into the submitted manuscript and resulted in the blurry of image. As for this problem, we amplified the image axis title font size and make them bold. If revision is acceptable, we will upload TIFF images with 600 dpi to the editorial department..

Point 3: The authors report the presence of citramalic acid as a metabolite contributing to the formation of betalains. Can the authors present a putative biosynthetic way for the formation of this metabolite and from this metabolite to the betalains.

Response 3: Information regarding the physiological function of citramalic acids in plants has been limited until now. We have made some efforts to fulfill the putative biosynthetic way between citramalic acid and betalain but failed to establish the putative biosynthetic way. We only supplemented limited physiological functions of citramalic acid in plants(The revised details can be found in Line 344-348, page 10-11.), Nevertheless your advice is very important and provide inspiration for us, we have found the shortcomings in current work and would follow your advice to improve my scientific research level.

Point 4: lines 101 and 111, the authors report that the color changing is due to a degradation of chlorophyll. Can the authors explain this degradation? What is the biosynthetic way between chlorophyll and betalains. Did the authors analyse the different chlorophyll a and b, but also xanthophylls.

Response 4: Peel color changed from pale green to reddish due to the decreased content in chlorophyll during pitaya fruit developing (Phebe et al, 2009, http://psasir.upm.edu.my/id/eprint/6716; Nerd and Mizrahi, 1997, http//:doi: 10.1002/9780470650608.ch7). About your suggestion, we supplemented the column chart of chlorophyll and carotenoid content (chart 1) of pitaya fruit during developmental stages (Firstly, freeze-dried samples were ground into powder, and then extracted with 80% methanol for 24h, next the absorption value was determined by ultraviolet spectrophotometer and the corresponding content was calculated). Compared with peel, the content of chlorophyll and carotenoid are trace in pulp,and the chlorophyll content in peel decreased significantly at 29DPA, which was consistent with the change trend of b* value in this paper. Indeed, Chlorophyll degradation and betalain formation were spatially successive in peel, and resulted in the color changing from green to red. However, there is yet no direct causal relationship between chlorophyll degradation and betalain synthesis, hence, the chart was not cited in paper. Thanks to your kind reminder, we have further verified the description in this paper to avoid mistakes.

Chart 1 The content of chlorophyll and carotenoid in pitaya fruit

Thank you for your suggestions, which are instructive to our thesis writing and scientific research work, and we learn more knowledge from your valuable comments.

With best wishes,

Yours sincerely,

Xiaopeng Wen

Institute of Agro-biotechnology,

Guizhou University, Guiyang 550025,

Guizhou Province, P R China
